# Relevance of BET Family Proteins in SARS-CoV-2 Infection

**DOI:** 10.3390/biom11081126

**Published:** 2021-07-30

**Authors:** Nieves Lara-Ureña, Mario García-Domínguez

**Affiliations:** Andalusian Centre for Molecular Biology and Regenerative Medicine (CABIMER), CSIC-Universidad de Sevilla-Universidad Pablo de Olavide, Av. Américo Vespucio 24, 41092 Seville, Spain; nieves.lara@cabimer.es

**Keywords:** SARS-CoV-2, COVID-19, BET, BET inhibitors, BRD4, BRD2, virus, immunity, inflammation

## Abstract

The recent pandemic we are experiencing caused by the coronavirus disease 2019 (COVID-19) has put the world’s population on the rack, with more than 191 million cases and more than 4.1 million deaths confirmed to date. This disease is caused by a new type of coronavirus, the severe acute respiratory syndrome coronavirus 2 (SARS-CoV-2). A massive proteomic analysis has revealed that one of the structural proteins of the virus, the E protein, interacts with BRD2 and BRD4 proteins of the Bromodomain and Extra Terminal domain (BET) family of proteins. BETs are essential to cell cycle progression, inflammation and immune response and have also been strongly associated with infection by different types of viruses. The fundamental role BET proteins play in transcription makes them appropriate targets for the propagation strategies of some viruses. Recognition of histone acetylation by BET bromodomains is essential for transcription control. The development of drugs mimicking acetyl groups, and thereby able to displace BET proteins from chromatin, has boosted interest on BETs as attractive targets for therapeutic intervention. The success of these drugs against a variety of diseases in cellular and animal models has been recently enlarged with promising results from SARS-CoV-2 infection studies.

## 1. Introduction

In late December 2019, a cluster of pneumonia cases caused by a new type of coronavirus appeared in Wuhan (capital of Hubei province, China). Chinese researchers sequenced the genome of this virus and the data were published on 9 January 2020. It was named Severe Acute Respiratory Syndrome Coronavirus 2 (SARS-CoV-2) and the disease caused by it, coronavirus disease 2019 (COVID-19) [1]. The spread of this virus was so rapid that on 30 January 2020, the World Health Organization (WHO) declared the disease a global public health problem, and on 11 March 2020, COVID-19 was classified by the WHO as a pandemic. According to daily updated data by WHO, more than 191,000,000 cases and more than 4,100,000 deaths have been confirmed by 22 July 2021 (https://covid19.who.int/, accessed on 22 July 2021).

Epidemiological studies have reported an incubation period of 1–14 days for SARS-CoV-2, with a peak of 3–7 days. During the latent period SARS-CoV-2 is highly contagious [2]. Infection in humans manifests from mild symptoms to severe respiratory failure (https://www.who.int/emergencies/diseases/novel-coronavirus-2019/question-and-answers-hub/q-a-detail/coronavirus-disease-covid-19, accessed on 22 July 2021). SARS-CoV-2 binds to epithelial cells in the respiratory tract and begins to replicate and migrate into the airways by entering the cells of the alveolar epithelium in the lungs. Replication of SARS-CoV-2 in the lungs may provoke a strong immune response, and the induced cytokine storm is associated with acute respiratory stress syndrome and respiratory failure, which is considered the leading cause of death in COVID-19 patients [3,4]. Patients older than 60 years and those with previous serious pathologies are at increased risk of developing acute respiratory stress syndrome and dying [5,6,7]. But besides respiratory breakdown, multiple organ failure has also been reported linked to COVID-19, affecting the heart [8], the liver [9], and the nervous system [10], among other organs [11,12,13]. In this context, current treatments, largely focused on alleviating symptoms, are not sufficient to efficiently control the infection. Thus, alternative strategies based on host-directed therapies through druggable targets are of high interest.

Recent reports on proteomic studies identifying host interactors for different proteins of SARS-CoV-2 opened new possibilities for treatment of COVID-19, as a number of these virus targets are druggable proteins [14]. Among them, members of the Bromodomain and Extra-Terminal Domain (BET) family of transcriptional coregulators stand out, since they are involved in activating a variety of relevant transcriptional programs in the cell. In this review, we update the recent discoveries linking BET proteins with SARS-CoV-2 and COVID-19, highlighting the promising results of treating SARS-CoV-2 infection with BET inhibitors.

## 2. BET Proteins

The BET family of proteins consists of a series of proteins that play an important role in gene transcription through epigenetic regulation, with a prominent impact in the control of cell growth and differentiation [15,16,17,18]. In mammals, the BET family is composed of four members: BRD2, BRD3, BRD4 and BRDT (Figure 1). While BRD2, BRD3 and BRD4 expression occurs ubiquitously, BRDT expression is restricted to the male germline [19]. Recently, BRD2 and BRD4 were reported to interact with the envelope (E) protein of SARS-CoV-2 [14], which makes of these BET members potential targets for host-directed therapy strategies. Moreover, several groups have demonstrated that angiotensin-converting enzyme 2 (ACE2), the main SARS-CoV-2 receptor for host cell entry, is under BET protein transcriptional regulation (see Section 4) [20,21,22,23].

A salient feature of this family of proteins is the presence of two tandem N-terminal bromodomains able to bind acetyl groups. Interaction with histones through acetyl-group recognition on lysine (K) residues constitutes the main mechanistic aspect of BET action. The tight relation of BET proteins with cell cycle progression explains why they are linked to many cancer types. This implication was initially illustrated in the context of fusions of BET members with the NUclear protein of the Testis (NUT), which gives rise to NUT midline carcinoma [16], but misregulation of BET expression is in the basis of many other types of cancer (reviewed in [24]). Thus, the development of drugs able to displace BET proteins from the chromatin as a therapeutic approach to fight cancer has been a highly active research objective in the last decade (reviewed in [25]). Besides, BET inhibition has also proven to be of interest for treating metabolic, cardiovascular, neurological and autoimmune diseases (reviewed in [26,27,28]). The inhibitory strategy has been long based on synthetic drugs mimicking the acetyl-lysine group, thus able to compete with chromatin for BET binding and to displace these from target sites. This approach successfully alleviates a number of cancer types in mouse and cellular models [29,30,31,32,33]. However, success in clinical trials with humans is limited, probably due to the toxicity of the high doses required for effective outcomes [25,34]. Of note, thrombocytopenia is among the most common and severe adverse events associated with BET inhibition [35]. Besides, despite the high selectivity of BET inhibitors for BET bromodomains, 44 additional human proteins have bromodomains [36], making it difficult to completely discount off-target effects.

Though BET inhibitors have limited success in the treatment of some cancers, recent reports have shown BET inhibition as a promising strategy for treating SARS-CoV-2 infection [20,21,22,23]. Thus, BET inhibition anticipates as a solid host-directed therapy against COVID-19. Strikingly, BET proteins are tightly linked to infection by other viruses. BET relation with viruses occurs at two levels. On the one hand, BET proteins are targets for several virus proteins, which may result in an impact on cell transcription [37]. On the other hand, BET proteins are involved in activating transcriptional programs related to immunity and the inflammatory response associated with infection [38,39].

### 2.1. BET Structure

BET family of proteins is characterized by a common domain structure (Figure 1), presenting from N- to C-terminus: the two bromodomains (BDI and BDII), a motif B, a well conserved extra-terminal (ET) domain and a less conserved domain called SEED due to the presence of serine and glutamic and aspartic acid residues [40]. Bromodomains comprise a conserved sequence of approximately 110 amino acids [41,42] that can bind to acetylated lysine residues in histones and other proteins, like the GATA-1 transcription factor [43,44,45]. These domains, through their interaction with nucleosomes in chromatin, are involved in epigenetic regulation of gene transcription [46]. Of note, BET bromodomains have been suggested to be the target of SARS-CoV-2 E protein [14]. Major histone recognition by BETs relies on acetylation of K5 and K12 on histone H4 [18,47,48,49,50,51,52,53,54]. Interestingly, SARS-CoV-2 non-structural protein (nsp) 5 interacts with histone deacetylase 2 (HDAC2) [14], leading to speculation about interference of SARS-CoV-2 with histone acetylation. The motif B presents a coiled coil structure that gives rise to an amphipathic helix and is an essential domain for homo- and heterodimerization of BET members, thus stabilizing its binding to the chromatin and facilitating its association to chromosomes during mitosis [55]. This region also contributes to partner recruitment [56,57]. The ET domain consists of a region of approximately 80 amino acids and is involved in the interaction with specific effector proteins [58,59]. Of note, the ET domain is the major target of most viral proteins interacting with members of the BET family [37] (Figure 1). For its part, both BRD4 and BRDT have a C-terminal domain (CTD) at their carboxyl extreme whose main function is to interact with the positive transcription elongation factor b (P-TEFb) [46,60,61]. Importantly, virus infection frequently associates with altered host transcription. In this context, the analysis of transcriptional profiles in cellular and animal models of SARS-CoV-2 infection and in patient samples, have permitted to identify unique transcriptional signature in response to the virus [62].

### 2.2. General BET Functions

A variety of reports link BET proteins with cell cycle progression. BRD2 interacts with and activates E2F, a transcription factor involved in the synthesis of proteins required for the G1 to S transition during the cell cycle [63,64], proving to be involved in the control of *Ccna2* (Cyclin A2) and *Ccnd1* (Cyclin D1) genes [50,65]. Initial reports on BRD4 suggested its participation in the G2 to M transition [66], but more recent reports have solidly established that it is required for the M to G1 transition [67,68]. On its hand, BRDT is required for expression of *Ccna1* (Cyclin A1) during spermatogenesis [69,70]. Importantly, SARS-CoV-2 nsp1, which is involved in inhibition of host protein expression, has been shown to induce cell cycle arrest in G0/G1 phase [71].

BRD4 was shown to be associated with chromosomes in interphase but also to remain bound to chromosomes during mitosis, when most nuclear regulatory factors are released into the cytoplasm [47,66]. Attachment to mitotic chromosomes has also been reported for BRD2 [55]. These observations, together with BET requirement for cell cycle progression, have led to consider BET proteins as true epigenetic factors, marking key chromatin positions from one generation to the next for timely activation of relevant cell cycle genes [72].

Several lines of evidence illustrate the role of BET proteins in organizing and maintaining the chromatin structure. BRDT has been shown to be a chromatin organizer in male germinal cells [17] and it was early observed that ectopic expression in somatic cells leads to dramatic reorganization of the chromatin [73]. In the same line, interfering with BRD4 leads to chromatin decondensation and fragmentation [74]. On its part BRD2 has been shown to cooperate with CTCF to enforce transcriptional and architectural boundaries at chromatin [75]. BET proteins act as histone chaperones of the acetylated nucleosomes they recognize, allowing the passage of RNA polymerase II to elongate nascent transcripts [50]. As mentioned, the CTD in BRD4 and BRDT is crucial for interaction with P-TEFb [46,60,61,76]. Interestingly, several viral proteins have been shown to compete with BRD4 for P-TEFb (see Section 3). BRD4 releases P-TEFb from inhibition by HEXIM1 [76]. Active P-TEFb phosphorylates Ser2 of the RNA Polymerase II C-terminal motif promoting RNA polymerase transcription elongation [77,78,79,80]. Recently, it has been described as a type of regulatory elements, the “super-enhancers” (SEs), which consist in large clusters of enhancers arranged in a cell type-specific manner [81,82]. They have emerged as key disease drivers when dysregulated, especially in the oncogenic transformation [83]. These elements represent a small fraction of the total enhancers in a cell, but they recruit a large proportion of regulatory proteins, strikingly BRD4, for the control of specific genes involved in the maintenance of cell identity. Thus, altered regulation of SEs may lead to changes in cell identity and cellular transformation. Indeed, cancer cells become highly dependent on these regulatory elements, so their specific targeting appears as a promising avenue for therapeutic intervention [84].

BET proteins are essential for development. Knock out mice for *Brd4* and *Brd2* die at early (postimplantation) and later (E11.5–E13.5) embryonic stages, respectively [85,86,87,88]. Heterozygous mice for these mutations also present defects, especially reduced cell growth. Particular association of BET proteins with development of the nervous system has been indicated. On the one hand, a BRD2 deficit is associated with defects in the developing neural tube, where the gene is highly expressed [85,87], and with a decrease in the number of GABAergic neurons [89]. On the other hand, BRD4 regulates the transcription of genes involved in synapses, enhancing learning and memory processes in mice [90]. On its hand, BRDT is well documented to play a prominent role in spermatogenesis [17,70].

## 3. BET Relation with Viruses

BET protein involvement in viral processes is not restricted to SARS-CoV-2. Abundant literature illustrates how BET proteins are strongly associated with infection by different types of viruses [37,39]. While some viral proteins directly interact with members of the BET family, others interfere with functional BET partners. The papillomaviruses E2 protein, the latency-associated nuclear antigen (LANA) of some herpesvirus, the integrase of some gamma-retroviruses and the E protein of SARS-CoV-2 are examples of viral proteins physically interacting with BET proteins [14,37], while human immunodeficiency virus type 1 (HIV-1) Tat or human T-lymphotropic virus type 1 (HTLV-1) Tax proteins can compete with BETs for PTEF-b [37]. Here, we summarize BET relation with different viruses, including SARS-CoV-2 (Figure 1 and Table 1), which outlines different strategies in viral infection leading to diverse scenarios for therapeutic BET inhibition.

The papillomaviruses E2 protein is essential for transcriptional activation and repression [112,113], for viral DNA replication in cooperation with the E1 protein [112] and for tethering of the viral genome to host mitotic chromosomes [114]. It was observed in bovine papillomavirus that BRD4 colocalizes with E2 in the mitotic chromatin and that E2 interact with the CTD of BRD4 [91]. BRD4 binding to E2 prevents E2 degradation, modulates E2-mediated transcription and tethers E2 to mitotic chromatin [92], despite some examples of E2 proteins associating with mitotic chromosomes in a BRD4 independent manner [115]. However, what has been undoubtedly shown is that BRD4 plays a dual role in regulating the transcriptional function of E2 proteins in papillomaviruses. Meanwhile different studies using diverse approaches show that BRD4 is essential for the activation of E2 transcriptional function [115,116,117,118,119], others show that BRD4 confers the ability to silence E2-mediated transcription [120,121]. Similarly, there are studies that attribute an essential role to BRD4 in viral genome replication [122], while others indicate that its role is not essential [123]. The most widely accepted current model contemplates that after infection of the cell by papillomavirus, BRD4 tethers the viral genome to active cellular chromatin to allow viral transcription. After binding to chromosomes, it recruits E1 and E2. As the genome replicates and the foci enlarge, BRD4 appears not to be required for continued replication of the genome (reviewed in [124]). More recent studies have shown that the interaction between the E2 protein and BRD4, besides occurring at the level of the CTD, also occurs at a basic residue-enriched interaction domain (BID). Moreover, it has been indicated that high-risk (HR) human papillomaviruses (HPV), associated with cervical cancer, but not low-risk (LR) HPV, associated with benign lesions of the genital tract [125], additionally interact with an N-terminal phosphorylation sites region (NPS) of BRD4 in a phosphorylation-dependent manner [93]. 

Kaposi’s sarcoma associated herpesvirus (KSHV), which causes Kaposi’s sarcoma, primary effusion lymphoma and some forms of Castleman’s disease, murine gamma-herpesvirus 68 (MHV-68) and the gamma-herpesvirus Epstein Barr virus (EBV), are also known to interact with BET proteins [99,100,126,127,128]. The LANA of the KSHV (kLANA) was the first viral protein discovered to interact with a member of the BET family protein [94]. kLANA plays important roles in replication [129,130,131,132,133], tethering of viral genome to cellular chromosomes [134,135,136,137,138,139,140] and regulation of viral and cellular gene transcription [129,130,141,142,143,144,145,146]. Its homologue in MHV-68, mLANA, is expressed in latency and during lytic replication and its function is essential in the establishment and maintenance of latency [147,148,149,150,151,152] and the EBV homologue, the Epstein-Barr virus nuclear antigen 1 (EBNA1), is involved in the regulation of viral transcription, replication and persistence [153]. Several studies show that kLANA and mLANA interact with a region containing the ET domain of BRD2, BRD3 and BRD4 [94,95,96,97,99], and a recent study has described that the preferential localization of kLANA and mLANA at transcription start sites (TSSs) [154,155,156,157] is due in part to BET proteins [98]. In this work, authors have shown that treatment with the BET inhibitor I-BET151 displaces kLANA protein, as well as BRD2 and BRD4, from viral and host TSSs. Mutations in mLANA preventing BRD2 and BRD4 binding [158] also have similar consequences [98]. In turn, EBNA1 has been described to interact with BRD4 and this interaction appears to play a critical role in EBNA1-mediated transcriptional activation, as it has been shown that *BRD4* silencing leads to a decrease in its transcriptional activity [100]. In addition, it has been reported that BET inhibition on the one hand prevents expression of the viral immediate-early protein BZLF1, but on the other hand it also prevents viral late gene expression, as BET members localize and act on lytic origins of replication [101]. A different relationship with BET proteins has been attributed to other herpesvirus, where no physical interaction occurs. This is the case for human cytomegalovirus (HCMV), being BET proteins pivotally connected to the regulation of cytomegalovirus latency and reactivation. During latency, P-TEFb remains sequestered due to its interaction with BRD4, which prevents transcription of viral genes. Treatment with BET inhibitors allows the release of P-TEFb and thus the transcription of HCMV lytic genes, thanks to its recruitment to the promoters of the super elongation complex. But all this occurs without inducing viral DNA replication and complete reactivation [111].

Regarding retroviruses, the murine leukemia virus (MLV) integrase is also known to interact with the ET domain of BRD2, BRD3 and BRD4 [102,103,104,105]. Similarly, the porcine endogenous retrovirus A/C (PERV A/C) integrase interacts with the ET domain of BET proteins. This integrase protein interaction appears to be gamma-retrovirus specific, as BRD2 does not interact with either Rous associated virus type 1 (RAV-1, alpha-retrovirus) or HIV-1 (lentivirus) integrases [106]. In the case of the HIV, BRD4 plays an important role in transcriptional regulation [60,107,108,159]. On the one hand, the Tat protein of HIV is essential for transcriptional elongation from the long terminal repeat (LTR) promoter and several studies have shown that BRD4 inhibits the transcriptional elongation by competing with Tat for cellular P-TEFb [60,107,108]. Therefore, inhibition of BRD4 expression would lead to increased Tat-mediated HIV gene transcription. But on the other hand, BRD4, through its bromodomains can also be recruited to the HIV LTR by interacting with acetylated histones H3 and H4. The effects on HIV transcription and latency establishment are different depending on which of the histones it interacts with [109]. Treatment with the BET inhibitors JQ1, apabetalone, PFI-1 or UMB-136, has proven to reactivate latent HIV, helping to eradicate the virus [107,108,160,161]. However, treatment with the small molecule ZL0580, has shown suppression of HIV induction by establishing a more repressive chromatin structure at the HIV LTR, as well as by inhibiting Tat-mediated transcription transactivation and elongation [162]. For its part, the retrovirus HTLV-1 encodes the Tax protein which plays an important role in viral replication, transformation and transcriptional activation [163,164,165,166,167,168,169]. Similar to HIV Tat, Tax protein competes with BRD4 for binding to P-TEFb, which is essential for LTR promoter transactivation by Tax. Therefore, HTLV-1 Tax protein, like HIV Tat protein, appears to mimic the function of BRD4 and competes with it for binding to P-TEFb [110]. Interestingly, inhibition of BRD4 with JQ1 results in impaired proliferation of Tax-positive HTLV-1-infected cells, and then, in reduced Tax-mediated cell transformation and tumorigenesis [170], suggesting that BET inhibitors could also be used as anti-cancer therapy in tumors caused by viral infections.

BRD4 inhibition leads to viral arrest in infection by different types of viruses. In cells infected with pseudorabies virus (PRV), herpes simplex virus type 1 (HSV1), ectromelia virus (ECTV), vesicular stomatitis virus (VSV), porcine reproductive and respiratory syndrome virus (PRRSV), Newcastle disease virus (NDV) and influenza virus (H1N1), inhibition of BRD4 with different drugs (JQ1, OTX-015 and I-BET151) has shown antiviral activity. In a number of cases no changes in viral transcription was observed upon BRD4 inhibition, but an attenuation of viral attachment [171].

It has also been shown that respiratory syncytial virus (RSV) infection affects the BRD4 interactome, increasing the interaction with transcription factors involved in the innate immune response and cellular stress, and that this recruitment of multiple transcription factors occurs in a manner dependent on acetyl-lysine recognition [172].

As mentioned, it was recently reported the interaction of the SARS-CoV-2 E protein with BRD2 and BRD4 [14]. In this work, it is indicated that the histone H2A N-terminal tail shares similarity with a region of about 15 amino acids of E protein. The indicated histone region contains the target K residues for acetylation recognized by BET members. This observation has led to suggest the involvement of the BET bromodomains in the interaction with E protein, and to speculate about E protein ability to disrupt BET interaction with chromatin, which may affect host transcription in benefit of the virus [14]. However, the main target domains for interaction with viral proteins on BET members are the ET and CTD domains. Thus, interaction of E protein with bromodomains is unexpected and needs confirmation.

## 4. BET Proteins and SARS-CoV-2 Infection

Viruses need to use the machinery of the cells they infect to replicate. As explained, first evidence of the SARS-CoV-2-BET relation resulted from a proteomic study by Gordon et al. revealing the interaction of the SARS-CoV-2 E protein with BRD2 and BRD4 [14]. As BET proteins are general transcriptional co-regulators, Gordon et al. suggested that BET interaction with E protein may cause gene expression changes in the host cell that could be beneficial to the virus cycle. Besides being involved in cell cycle progression, BET proteins are also relevant for regulation of immunity and inflammation. Indeed, one decade ago, the pioneering BET drug I-BET was shown to displace BET proteins from regulatory elements on key inflammatory genes, displaying then anti-inflammatory properties [173]. Recent works have revealed that the beneficial use of BET inhibitors against SARS-CoV-2 is beyond the BET-E protein interaction.

Detection of viral antigens leads to antigen presentation to natural killer cells and CD8^+^ T-cells, which activate the immune response by production of proinflammatory cytokines and chemokines [174]. Normally, viruses are first faced by the innate immune response, but efficient counteraction may also require mobilization of the adaptive immune response (antigen targeting by immune cells). SARS-CoV-2 infection is linked to both responses [175]. However, BETs, and especially BRD4, are mainly involved in the innate immune response (reviewed in [39]). Production of proinflammatory cytokines is highly dependent on the Nuclear Factor kappa-light-chain-enhancer of activated B cells (NF-ĸB) signaling pathway [176]. Cytokines are essential for the immune response, but their aberrant dysregulation leads to hyperinflammation and may cause severe damage to tissues resulting in organ failure and death. This uncontrolled systemic inflammatory response is known as cytokine storm (CS) [177] and may account for up to 5% of COVID-19 patients [12]. Indeed, over-stimulation of the immune response can be more damaging than the virus infection itself. ACE2 is involved in cleaving angiotensin II into anti-inflammatory angiotensins 1–7. Thus, it has been suggested that ACE2 blocking by SARS-CoV-2 binding should result in accumulation of pro-inflammatory angiotensin II, contributing to the exacerbated immune response [178]. However, uncontrolled inflammation might also rely on pre-inflammatory states of certain organs and/or tissues.

### 4.1. SARS-CoV-2

Coronaviruses belong to the subfamily *Orthocoronavirinae* of the family *Coronaviridae* in the order *Nidovirales*. They are highly diverse, enveloped, positive-sense and single-stranded RNA viruses [179]. According to their genome structure and phylogenetic relationships, there are four genera of coronaviruses: alpha-, beta-, gamma- and delta-coronaviruses. While gamma- and delta-coronaviruses infect birds and some mammals, alpha- and beta-coronaviruses are responsible for the infection of various types of mammals [180], causing respiratory disorders in humans and gastroenteritis in other animals [181,182]. SARS-CoV-2 belongs to the beta-coronaviruses and is the seventh member of the coronavirus family to cause infections in humans [183]. Among coronaviruses infecting humans we find several common cold viruses like HCoV-OC43, HCoV-HKU1 and HCoV-229E [184], but coronaviruses with a high pathogenic capacity in humans have emerged in the last two decades, including SARS-CoV in 2002 and 2003 which caused 8000 confirmed cases with a 10% fatality rate, and MERS-CoV in 2012 with 2500 cases and a death rate of 36% [185]. Analysis of SARS-CoV-2 genome revealed 79.5% genomic identity with SARS-CoV [182].

The SARS-CoV-2 genome of approximately 30 kb encodes 14 open reading frames (ORFs) (Figure 2A). The ORF1ab is located in the 5′ region and encodes the overlapping polyproteins 1ab and 1a, which undergo auto-proteolysis to give rise to 16 non-structural proteins (nsps), mostly contributing to the formation of the replicase/transcriptase complex (RTC). At the 3′ end up to 13 ORFs are present, which include four structural proteins (Figure 2B): Spike (S), Envelope (E), Membrane (M) and Nucleocapsid (N) and nine putative accessory factors [186,187].

The N protein binds to the viral genome and is involved in RNA replication, virion formation and immune evasion, and also interacts with the M protein [188]. The M protein promotes viral particle assembly and budding through interaction with the N protein and accessory proteins 3a and 7a [174,189]. The most relevant structural proteins for BET inhibition are E and S. E protein has been demonstrated to interact with BRD2 and BRD4 [14]. It is the smallest structural protein and facilitates the production, maturation and release of virions [190]. S protein is a transmembrane protein that facilitates the binding of the viral envelope to the host receptor. The main receptor of SARS-CoV-2 in host cells is ACE2 [182,191], which is expressed in the colon, gallbladder, heart, kidney, epididymis, breast, ovary, lung, prostate, esophagus, tongue, liver, pancreas, and cerebellum [192]. Similar to other coronaviruses, SARS-CoV-2 needs to proteolyze S protein to activate the endocytic pathway. Host proteases, including transmembrane protease serine 2 (TMPRSS2), cathepsin L and furine have been shown to be involved in the process [191,193,194]. TMPRSS2 is highly expressed in certain tissues and co-expressed with ACE2 in bronchial branches, lungs and nasal epithelial cells, which explains part of the tissue tropism of SARS-CoV-2 [195,196]. Notably, expression of ACE2, but also of TMPRSS2, is under the control of BET proteins, what has been exploited to fight SARS-CoV-2 infection [20,21,22,23] (see below).

Of the 4 structural proteins, SARS-CoV-2 shares more than 90% amino acid identity with SARS-CoV, except for the S protein, which presents greater divergence [182,197]. SARS-CoV uses the same receptor as SARS-CoV-2 to infect cells, suggesting that both viruses may have similar life cycles [182]. However, the binding affinity of the S protein of SARS-CoV-2 to the human receptor is much higher than that of SARS-CoV [198]. This may be due to sequence differences, which makes the binding affinity of SARS-CoV-2 to ACE2 10 to 20 times higher than that of SARS-CoV, thus, enhancing the spread of SARS-CoV-2 [199,200]. The uncontrolled expansion of the virus is giving rise to new variants, with enhanced infectivity in some cases, that can challenge the control of the pandemic and compromise the efficiency of recently developed vaccines. Both, mutations in S protein leading to enhanced affinity for ACE2 receptor, and mutations reducing neutralizing activity of antibodies (immune escape), may be related to higher infectivity of new variants [201]. Since all SARS-CoV-2 variants use ACE2 for entry into the host cell, it is predicted that strategies targeting ACE2, should be effective in reducing infection by new variants. These strategies include BET inhibition, as explained below.

Once SARS-CoV-2 has entered the cell, the genetic material is released into the cytoplasm and starts translation. The first and only region directly translated from the genome is that coding for nsps (ORF1ab) [202]. Polyproteins 1ab and 1a are processed by the action of chymotrypsin-like protease (3CLpro, or main protease (Mpro)) and one or two papain-like proteases (PLpro) that are encoded by the virus [203]. The RTC locates in double membrane vesicles creating a protective microenvironment for replication of genomic RNA and transcription of subgenomic mRNAs. The subgenomic mRNAs are translated into accessory and structural proteins M, S and E that are isolated in the endoplasmic reticulum and then translocated to the endoplasmic reticulum-Golgi intermediate compartment. Subsequently they interact with the newly produced genomic RNA encapsidated by N protein, resulting in the formation of vesicles that are exported out of the cell through exocytosis [202,204,205].

### 4.2. SARS-CoV-2 Induced Immune Response and BET Proteins

The immune response is activated through recognition of both pathogen-associated and damage-associated molecular patterns by cell surface and intracellular pattern recognition receptors. In this scenario, toll-like family of pattern recognition receptors (TLR) play an important function. Expression of both TLR3 and TLR4 is upregulated by SARS-CoV-2 [206,207]. In addition, S protein interacts with and activates TLR4 [208,209]. BETs have been shown to positively regulate TLR4 expression in pancreatic ductal adenocarcinoma and in acute myocardial infarction rodent models [210,211]. In turn, TLR3 is activated by foreign RNAs molecules (particularly dsRNAs derived from virus replication), and it has been established that TLR3-induced acute airway inflammation and remodeling is efficiently neutralized by BET inhibitors, as it depends on BRD4 [212].

TLR3 and TLR4 stimulation results in NF-ĸB activation [174]. Of note, SARS-CoV-2 infection results in higher NF-ĸB pathway activation [213]. Notably, BRD4 control of innate immunity largely relies on BRD4 regulation of canonical NF-ĸB pathway, associated with transcription factor RELA. Under normal conditions, IĸBα blocks RELA in the cytoplasm. Inflammation-associated activation of IĸB kinases results in IĸBα phosphorylation, which is ubiquitinated and then targeted for degradation. Liberated RELA, undergoes translocation to the nucleus for regulation of inflammatory and immunomodulatory genes. RELA-mediated transcription activation depends on BRD4. Transcription activation by RELA requires its acetylation at K310, which is recognized by BRD4 [214]. BRD4 binding seems to stabilize RELA, since BET inhibition or BRD4 depletion leads to RELA ubiquitination and degradation [214,215]. In turn, RELA acetylation requires phosphorylation at S276 [216], which also depends on BRD4 [217]. Phosphorylation-coupled acetylation of RELA has been shown to facilitate BRD4 binding and recruitment of P-TEFb for transcriptional elongation of inflammatory cytokine genes upon RSV viral infection [216].

Activated NF-ĸB cooperates with Interferon Regulatory Factor (IRF) 3 to induce proinflammatory cytokines like type I interferon molecules (IFNs) and Tumor Necrosis Factor (TNF) [174]. Notably, upregulation of both types of molecules has been observed in COVID-19 patients [207]. IRFs are the main transcription factors involved in production of IFNs and are key regulators of antiviral immunity. It has been described that following RSV viral infection, the BRD4/RELA complex recruits the P-TEFb component CDK9 to *IRF1* and *IRF7* promoters for enhanced expression, and BRD4 inhibition has proven to alleviate viral-associated inflammation in this system [212]. Besides, it has been shown that virus infection in macrophages downregulates BRD3 expression and that BRD3 depletion impairs virus-mediated production of IFN-ß [218].

The tumorigenesis-associated JAK-STAT pathway also cooperates with NF-ĸB in triggering the immune response [219]. Through extracellular stimuli (mainly interleukin-6), the membrane receptor-associated Janus kinase (JAK) activates Signal Transducer and Activator of Transcription (STAT) factors to regulate the expression of cytokine-responsive genes [220]. JAK-STAT activation results in phosphorylated STAT, which enters the nucleus to activate transcription of IFN-stimulated genes. In human pluripotent stem cell-derived cardiac organoids (hPSC-COs), Mills et al. have reported that simulation of the COVID-19-associated CS leads to phosphorylation of STAT1 at S727 site [21]. On the other hand, BET inhibition has shown to efficiently inhibit the phosphorylation of STAT3 [221]. Moreover, combined inhibition of BET and JAK proteins has been shown to efficiently reverse inflammation linked to bone marrow fibrosis [222]. *Brd4* siRNA delivery through liposome nanoparticles efficiently suppresses RELA and STAT3 activation in LPS-induced mouse models of inflammation [223].

Another important determinant of the immune response triggered by SARS-CoV-2 is NLRP3, which is well expressed in various cell types such as lung epithelial, kidney, cardiac, endothelial, hematopoietic and innate immune cells [224]. NLRP3 is the most studied component of the inflammosomes, which are multiprotein oligomers of the innate immune system responsible for the activation of the inflammatory response. Inflammosomes strikingly participate in Caspase 1 activation, which leads to the induction of pyroptosis, a newly introduced type of programmed cell death associated with inflammation [225]. It has been shown that BRD4 inhibition prevents proliferation and epithelial to mesenchymal transition in renal cell carcinoma by increasing NLRP3 levels, which results in activated Caspase 1 and pyroptosis [226].

Besides triggering inflammation, SARS-CoV-2 infection, like other respiratory viral infections, is linked to oxidative stress of the epithelium, whose cells activate the transcription factor Nuclear factor erythroid-derived 2-Related Factor 2 (NRF2) for protection against oxidation and inflammation [227,228]. Notably, it has been described that BRD4 downregulation or BET inhibitors lead to NRF2 stabilization, which results in decreased reactive oxygen species production [229].

TNF is a well-known mediator of inflammation-associated heart failure, which induces systolic dysfunction [230]. Indeed, simulated SARS-CoV-2 infection by TNF treatment of hPSC-COs also leads to systolic dysfunction [21]. Although quite different in origin, atherosclerosis and heart failure have in common the involvement of TNF in mediating the inflammatory response. It has been shown that TNF-mediated inflammation in endothelial cells directs the formation of RELA- and BRD4-dependent SEs, being BET inhibition able to abrogate SEs-derived transcription and atherosclerosis [231]. These types of SEs are regulated in a highly dynamic way. They are tightly associated with disease [27] and have been denominated “latent enhancers”, being usually formed in response to noxious stimuli in terminally differentiated cells [232]. In these conditions they are flooded with transcription-associated proteins, sharing many features with SEs. Once stimulus ceases, most of them remain in a latent state of memory, which enables faster and greater induction by next stimulation. Cardiac hypertrophy promotes great changes in methylation and acetylation of chromatin leading to activation of a great number of enhancers [233]. In heart failure models, BRD4 occupies the majority of activated enhancers and BET inhibition disturbs associated transcription, suppressing cardiomyocyte hypertrophy [234].

### 4.3. BET Inhibition for COVID-19 Treatment

Since organ damage associated with SARS-CoV-2 infection is tightly linked to overactivation of the immune response, a way to avoid damage is to neutralize hyperinflammation. On top of that, blocking virus recognition and entry into host cells will prevent triggering of the immune response, which will also result in aborted inflammation. BET inhibition has a lot to do with both approaches. We have highlighted the fundamental role BET proteins play in the control of the immune response, and we have indicated that expression of the main receptor in host cells for SARS-CoV-2 entry, the ACE2 protein, but also that of the associated TMPRSS2 protease, is under the control of BETs. Thus, BET inhibition, by interfering at different levels of the virus infection, may result in beneficial outcomes when used for treating COVID-19. It has been indicated that ACE2 expression is stimulated by activation of the immune response [235], and early in SARS-CoV-2 research became clear that antagonizing ACE2 and/or TMPRSS2 could be of interest for COVID-19 therapies [191,236,237,238,239].

From an unbiased CRISPRi screen to uncover druggable pathways controlling SARS-CoV-2 S protein binding to human cells, Tian et al. have determined that BRD2 is a key player of the cellular response to SARS-CoV-2 infection [23]. In this screen, they used Calu-3 cells, a lung epithelial cancer cell line that endogenously expresses ACE2. As expected, *ACE2* downregulation was the major cause of impaired S protein binding. However, they also found *BRD2* among downregulated genes leading to decreased S protein binding. In fact, they showed that downregulation of *BRD2* correlated with lower levels of *ACE2* transcript and thereby of protein. Overexpression of the full BRD2 protein recovered *ACE2* transcriptional levels, demonstrating that BRD2 is required for ACE2 expression. Furthermore, it was shown that downregulation of *BRD2* produced a complete inhibition of viral replication in these cells, showing levels similar to those observed when *ACE2* was downregulated. The use of BET inhibitors produced effects similar to those observed upon *BRD2* downregulation, with reduced *ACE2* mRNA levels and S protein binding. Decreased *ACE2* mRNA levels was observed in both primary human bronchial epithelial cells and cardiomyocytes. Viral replication was also affected by BET inhibitors, and in a similar way to that observed when downregulating *BRD2* or *ACE2*. In addition, BET inhibition led to marked downregulation of genes that are involved in the response to type I IFN, whose expression is induced by SARS-CoV-2 both in patients and in cell cultures [23]. Therefore, these results suggest that BRD2 could be used as a therapeutic target for the treatment of COVID-19.

Moreover, SARS-CoV-2 infection is known to cause cardiac damage and dysfunction in 20–30% of hospitalized patients [240] and in the absence of infection well known inflammatory mediators such as TNF are associated with heart failure [230]. Mills et al. used hPSC-COs models, phosphoproteomic studies and single nuclei RNA sequencing to identify therapeutic targets and treatments for cardiac dysfunction. They studied the effects of several proinflammatory cytokines that are increased in COVID-19 patients and observed that they produced cardiac dysfunction, being TNF associated with systolic dysfunction and combination of IFN-γ, IL-1β and poly(I:C) with diastolic dysfunction, which is one of the most common dysfunction observed in COVID-19 patients [241]. The cardiac CS produced by IFN-γ, IL-1β and poly(I:C) induced 91 phosphosites, including one site on STAT1 and two sites on BRD4. Specific inhibitors exist for both proteins. However, while the different treatments used to inhibit STAT1 phosphorylation did not prevent CS-induced diastolic dysfunction, of several BET inhibitors tested (INCB054329, JQ1, RXV-2157, apabetalone and ABBV-744), the first four showed protection. This study demonstrated that CS-mediated diastolic dysfunction is mediated by BRD4-dependent mechanisms that can be blocked using BET inhibitors. Also in mice, they showed that response in the heart triggered by SARS-CoV-2 infection was partially blocked by treatment with INCB054329. Pre-incubation of hPSC-COs with INCB054329 prior to infection reduced ACE2 expression and decreased intracellular viral RNA, demonstrating the potential of BET protein inhibitors to block SARS-CoV-2 infection and prevent dysfunction [21]. BET inhibitors with dual BDI and BDII activities may display side effects [34], making necessary to determine the selectivity of the inhibition. Importantly, molecules specifically inhibiting BDII, like RXV-2157 and apabetalone, efficiently blocked SARS-CoV-2 infection by decreasing ACE2 expression and thereby SARS-CoV-2 S protein binding, showing that selective inhibitors against BDII are potential candidates to prevent heart damage caused by COVID-19 [21].

In line with this, a more recent study carried out by Gilham et al., shows that apabetalone, as JQ1, produces downregulation of *ACE2* expression in different cell types: Calu-3 cells, Vero E6 cell (monkey kidney epithelial cells), hepatocarcinoma cells HepG2 and Huh-7, and primary human hepatocytes [20]. Decrease in mRNA levels is accompanied by a decrease in protein levels in Calu-3 and Vero E6 cells. Apabetalone treatment also decreased *DPP4* expression in Calu-3 cells. *DDP4* encodes for Dipeptidyl peptidase 4 (CD26), a potential cofactor for SARS-CoV-2 entry into the host cell, as its presence on the cell surface facilitates viral binding [242,243]. Apabetalone and other BET protein inhibitors attenuate SARS-CoV-2 S protein binding and abrogate SARS-CoV-2 infection. These results and its well-established safety profile, together with the dual mechanism of action simultaneously combating hyperinflammation [244,245,246,247] and ACE2-mediated viral entry, make apabetalone a good candidate for treating SARS-CoV-2 infection [20]. Indeed, a clinical trial with apabetalone for COVID-19 treatment has been approved: NCT04894266 identifier (https://clinicaltrials.gov/, accessed on 22 July 2021).

Additional studies have reinforced the interest on BET targeting as an effective tool against SARS-CoV-2 infection. Expression of the key host proteins ACE2 and TMPRSS2 mediating virus entry in the cell is regulated by androgens [22,248] and transcriptional repression of the androgen receptor (AR) enhanceosome with AR or BET inhibitors suppressing SARS-CoV-2 infection in vitro [22]. Studies have shown that *AR, ACE2* and *TMPRSS2* are co-expressed in various types of human and murine lung epithelial cells, including alveolar and bronchial cells. Adult immune-competent C57BL/6 male mice were castrated to create an androgen-deprived condition and compared with non-castrated mice and with castrated mice treated for 5 days with testosterone. The experiments showed that *Tmprss2* and *Ace2* are positively regulated by androgens [22]. In the AR-positive LNCaP prostate cancer cells, which can be infected by SARS-CoV-2, it has been shown that blocking AR signaling with the use of different AR antagonists approved for prostate cancer treatment affects the infective capacity of SARS-CoV-2, showing a dose-dependent decrease in the expression of *TMPRSS2* and *ACE2*. Likewise, the use of different BET protein inhibitors showed decreased expression of *TMPRSS2* and *ACE2*, suggesting once again that BET proteins play a role in regulating the expression of the SARS-CoV-2 entry factor ACE2 and that this is independent of AR regulation, supporting the idea that inhibition of BET proteins may be useful to mitigate SARS-CoV-2 infection [22].

SUPT16H protein together with SSRP1 constitute FACT (FAcilitates Chromatin Transcription), a heterodimeric histone chaperone associated with chromatin remodeling during gene transcription [249]. BRD4 stabilizes SUPT16H by recognizing SUPT16H acetylation at K647 [250]. Targeting SUPT16H by RNAi-based approaches or pharmacological inhibition leads to the induction of interferons and interferon-stimulated genes, efficiently inhibiting SARS-CoV-2 infection, but also infection by other viruses like Zika and influenza [250]. This raises the question whether the effects of BET inhibitors on SARS-CoV-2 infection are mediated, at least in part, by altered SUPT16H stability. Besides BET proteins, another critical druggable target identified in SARS-CoV-2 interactome is mTOR [14]. It has been shown that the dual BET/mTOR-PI3K-α SF2523 inhibitor effectively blocks SARS-CoV-2 replication in lung bronchial epithelial cells in vitro [251]. Moreover, synergistic effects are observed when SF2523 is combined with remdesivir. Additional molecules like flavonoids and allergen fragrance molecules have been proposed as compounds of interest to interfere with SARS-CoV-2 infection through BET inhibition [252,253].

In sum, BET inhibition for beneficial effects on SARS-CoV-2 infection seems to operate at different levels (Figure 3). By targeting BETs, different processes/pathways relevant for virus infection can be simultaneously interfered. As explained, BET inhibition should result in attenuated inflammation but also in decreased ACE2 expression, which will result in hampered infection, collectively leading to reduced tissue damage. Besides, direct targeting of host proteins by viral proteins may also have consequences but it is an additional piece of the landscape at best. In the case of E protein interacting with BETs we do not know at present whether interaction leads to significant impaired BET function. This seems not to be the case, as Tian et al. have reported that E protein overexpression has mild and non-overlapping effects on transcriptome in comparison with the effect of *BRD2* knockdown or BET inhibition [23]. In the case that the bromodomains mediate the interaction with E protein we can anticipate that the use of BET inhibitors may be detrimental for the cell, as inhibitors can enhance E protein-mediated interference at bromodomain level. However, we can speculate about the use of well-defined concentrations of BET inhibitors resulting in E protein dissociation without grossly affecting chromatin attachment of BET proteins, thus resulting in a benefit for the cell. Nevertheless, as demonstrated, the use of BET inhibitors has proven to be an efficient tool in fighting SARS-CoV-2 infection effects for many other reasons.

## 5. Conclusions

BET proteins appear as master transcriptional coregulators of many and essential cellular processes. Several proteins from different viruses have been revealed to interact with BET proteins, but BETs are also key regulators of innate immune response and thereby of inflammation associated with viral infection. Inhibition of BETs has proven to efficiently fight different inflammation-associated diseases and viral infection processes, in cellular and animal models. In particular for cancer treatment, great efforts have been made to translate these results into the clinic. In this field, unfortunately, clinical trials have not yielded the desired results, due to toxicity of the elevated doses required for efficient cancer arrest. However, BET inhibitors may prove to be effective at non-toxic concentrations for many other BET-linked diseases. Recent determination of the SARS-CoV-2-associated proteome has revealed the presence of several druggable targets, among them BET proteins. Moreover, recent works have proven efficient reduction of SARS-CoV-2-associated noxious effects by BET inhibitors, opening new perspectives for host-directed therapeutic intervention against COVID-19. This ultimately is need of clinical trials, and not exclusively focused on BET inhibitors. The combined use of BET inhibitors with other drugs is emerging as a promising tool for efficient treatment of inflammation-associated diseases.

## Figures and Tables

**Figure 1 biomolecules-11-01126-f001:**
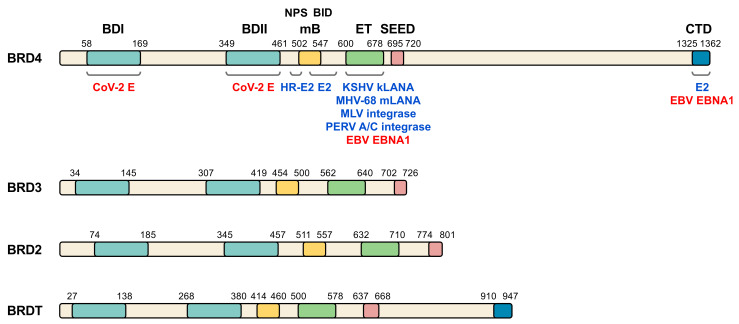
BET family of proteins and viral interacting proteins. The four members of human BET family are represented and amino acid position of the main relevant domains is indicated (BDI, bromodomain I; BDII, bromodomain II; mB, motif B; ET, extra terminal domain; SEED, SEED domain; CTD, C-terminal domain; NPS, N-terminal phosphorylation sites region; BID, basic residue-enriched interaction domain). Different virus proteins demonstrated to directly interact with BET domains (blue) or postulated to interact (red) are shown on BRD4, although some of the indicated proteins also interact with other BET members. CoV-2 E, SARS-CoV-2 E protein. See Table 1 and text for details and references.

**Figure 2 biomolecules-11-01126-f002:**
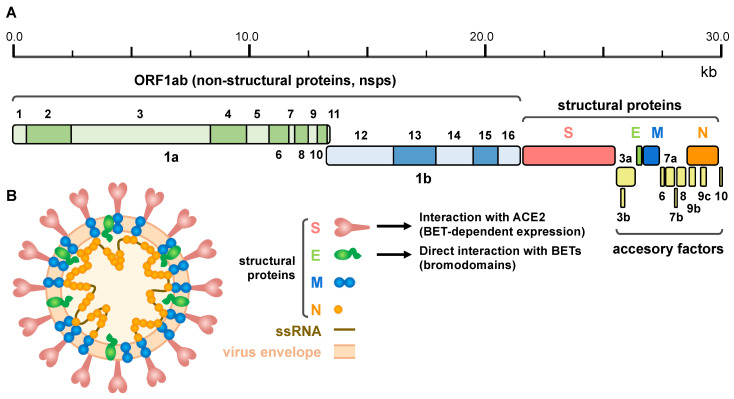
SARS-CoV-2 genomic structure and proteins. (**A**) Schematic representation of the SARS-CoV-2 genome, indicating the different derived proteins. ORF 1ab give rise to two polyproteins for non-structural proteins: pp1a and pp1ab, this last through a programmed ribosomal frameshift. Structural proteins comprise: Spike (S), Envelope (E), Membrane (M) and Nucleocapsid (N) proteins. (**B**) Schematic representation of the viral particle showing the viral envelope, the single strand RNA (ssRNA) genome and structural proteins (proteins are not represented to scale). Interaction of S protein with ACE2 (under the control of BET proteins) and direct interaction of E protein with BETs (putatively involving the bromodomains) is also indicated.

**Figure 3 biomolecules-11-01126-f003:**
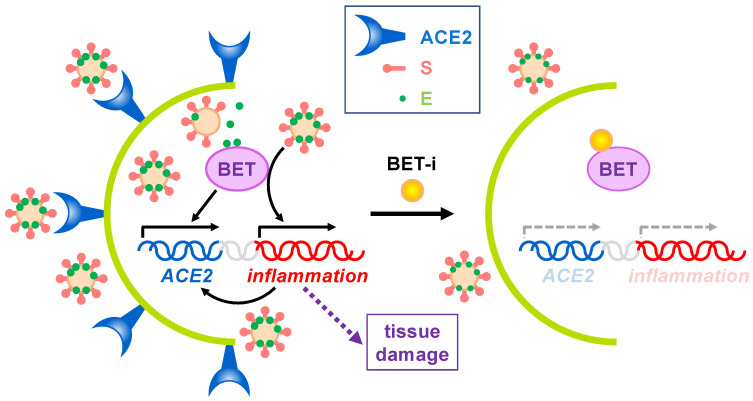
Effects of BET inhibition on SARS-CoV-2 infection. On one hand, expression of the SARS-CoV-2 receptor ACE2 depends on BET proteins. On the other hand, virus infection triggers the immune response, leading to BET-dependent activation of inflammation, which in turn may also activate *ACE2* expression. Uncontrolled inflammation may cause severe tissue damage. Besides, SARS-CoV-2 E protein interacts with BETs, but no associated effects have been reported to date. The use of BET inhibitors (BET-i) attenuates ACE2 expression and counteracts inflammation, thus, reducing infection and tissue damage.

**Table 1 biomolecules-11-01126-t001:** BET interactions with viruses.

Virus	Viral Protein/Genome	BET	BET Domain	Functions	Refs.
Papillomavirus HR-HPV	E2 HR-E2	BRD4	CTD, BID CTD, BID, NPS	E2 stability, E2-mediated transcription, E2 tethering to mitotic chromatin	[91,92,93]
KSHV	kLANA	BRD2, BRD3, BRD4	ET	kLANA tethering to chromatin and TSSs	[94,95,96,97,98]
MHV-68	mLANA	BRD2, BRD3, BRD4	ET	mLANA tethering to chromatin and TSSs	[98,99]
EBV	EBNA1 *OriLyt* **	BRD2, BRD3, BRD4	(ET, CTD) * (BDs)	EBNA1-mediated transcription Late gene expression	[100,101]
MLV	integrase	BRD2, BRD3, BRD4	ET	Integration into TSSs and CpG islands	[102,103,104,105]
PERV A/C	integrase	BRD2, BRD3, BRD4	ET	Integration cofactor	[106]
HIV	Tat *** *LTR*	BRD4	CTD BDs	Competence for P-TEFb HIV transcription and latency	[60,107,108,109]
HTLV-1	Tax	BRD4	CTD	Competence for P-TEFb	[110]
HCMV	*Promoters*	BRD4	CTD	Competence for P-TEFb for transcription	[111]
SARS-CoV-2	E	BRD2, BRD4	(BDs)	(Transcription)	[14]

* brackets indicate putative interaction domains or function. ** italic indicates virus genome regions. *** underlining indicates non-direct interaction but competition for P-TEFb.

## Data Availability

Not applicable.

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
