# Peer review of "Relevance of BET Family Proteins in SARS-CoV-2 Infection"

_biomolecules, 2021, doi:10.3390/biom11081126_

Round 1

Reviewer 1 Report

In this manuscript, Dr. Urena and Dr. Garcia-Dominguez discuss the most recent molecular evidence supporting the use of BET inhibitors in the treatment of SARS-CoV-2 infection.

SARS-CoV-2 infection is a world-wide emergency and new perspective to increase treatment options is hot-topic.

The provided information are clear and concise. The entire manuscript results well-organized and readable and thus, very interesting.

Minor comments

- line 113: please spell/clarify the term nsps

- lines 187-193: these concepts should be reorganized to give information in a more concise and clear manner.

-line 255: define NUT proteins

Author Response

We thank the reviewer for the positive assessment. Now NUT term has been clarified/defined in the required place. nsps has been kept without spelling, since this abbreviature was defined 3 paragraphs before. Concepts formerly in lines 187-193 have been shortened and reorganized.

Reviewer 2 Report

The manuscript by Nieves Lara-Ureña and Mario García-Domínguez “Relevance of BET family proteins in SARS-CoV-2 infection” reviews BET protein involvement in SARS-CoV-2 infection and states the following objective: “In this review we update the recent discoveries linking BET proteins with SARS-CoV-2 and the COVID-19.”

Overall, the manuscript is well written and emphasizes how BET proteins play roles in multiple and distinct biological processes.  However, as outlined below, the authors do not connect large sections of the manuscript to the title and objectives of this review.  Further, critically important concepts are either missing, or lack sufficient detail to summarize how BET proteins are involved in SARS-CoV-2 infection or manifestation of severe symptoms.  Finally, there are discrepancies between text and figures that the authors should be aware of.

Major comments:

Section 3 is a large part of the manuscript describing BET protein structure – function and diverse roles of BET proteins in health and disease.  The section is accurate, but fails to connect to SARS-CoV-2.  It could be the basis of a different review in the BET protein field.  It should also be noted that Section 3.3: “BET proteins in cancer and other diseases” not only deviates from the stated objective of this review, but is a dramatically curtailed summary of an extensively examined field. 

Section 4 is also a large section of the manuscript.  Here BET protein involvement in the life cycle of viruses other than SARS-CoV-2 is outlined. The processes in this section are not clearly related to (a) coronaviruses or (b) SARS-CoV-2.  Like section 3, section 4 could be the basis of a different review that’s not limited to SARS-CoV-2.  Together, Sections 3-4 represent a substantial part of the manuscript which is apparently off topic.

Section 5 promises to connect BET proteins and BET inhibition with COVID-19.

  • Section 5.1 does not discuss COVID-19. Rather, it outlines BET protein involvement in inflammatory processes at the molecular level, which is inconsistent with the section title. Cancer, fibrosis, atherosclerosis, and cardiac hypertrophy are discussed, but SARS-CoV-2 is not (it should be). 
  • The first paragraph in Section 5.2 describes how BET inhibitors suppress inflammatory processes. The authors make reference to models of the cytokine storm in cardiac organoids and in LPS treated mice in studies by Mills et al (reference 214), however, this section could tie specific molecular processes in Section 5.1 with the COVID-19 cytokine storm.  This comes across as an oversight.  BET inhibitors could suppress the cytokine storm underlies mortality, or the tissue damage associated with “long COVID”.  So, the topic of suppressing SARS-CoV-2 induced inflammation deserves more regard.
  • Several groups have independently shown that BET inhibitors dramatically reduce SARS-CoV-2 infection of human cells in vitro (reference 213, 214, 216, doi.org/10.3390/biomedicines9040437). This is briefly mentioned in section 5.2, and deserves more focus.  Importantly, these publications relate reduced SARS-CoV-2 infection with BET inhibitors suppressing ACE2 gene expression – see below. 
  • ACE2 is required for entry of SARS-CoV-2 into host cells. However, the authors do not connect cell types expressing ACE2 with cell types / organs they list as damaged by SARS-CoV-2 in the 3rd paragraph of Section 1.  Therefore, the manuscript fails to emphasize how BET inhibition may reduce damage in multiple organs that are susceptible to SARS-CoV-2 infection.  Same for TMPRSS2 (reference 216).
  • There is no indication of a clinical trial using a BET inhibitor to treat COVID-19 in this manuscript (ClinicalTrials.gov Identifier: NCT04894266).  This seems fundamental for a section titled, "BET inhibition for COVID-19 treatment".

Discrepancies between text and figures authors should be aware of:

  • Section 2.1 states that the SARS-CoV-2 genome contains 14 open reading frames, but 14 open reading frames isn’t clear in figure 1A.
  • Figure 1B has two elements named Membrane: Membrane protein and Membrane lipids. This should be clarified.
  • Table 1 states SARS-CoV-2 E-protein binds to BET bromodomains, but Figure 2 shows E-Protein binds between bromodomains. This is an apparent contradiction. 

Author Response

Regarding section 3, we understand the point of the Reviewer. Nevertheless, we think it is important to keep such a section in order to give an idea about classic and general functions of BETs, as we think it is relevant to understand why BET inhibitors have the potential to fight SARS-CoV-2 infection. However, we have shortened and reorganized section 3, now section 2, and we make mention to SARS-CoV-2 at several points. Section 3.3 has been now removed.

Regarding section 4, it is true, as Reviewer indicates, that this section is not largely related to SARS-CoV-2, but we think it is important to keep in mind that before being related to SARS-CoV-2, BET proteins were related to many other viruses. We think this is an opportunity to update knowledge on BETs and viruses, because the way they interact with classic viruses may be useful to understand the ways they interact with new viruses, and how to exploit it to fight them. Nonetheless, we have shortened this section (now section 3) and included in it the section formerly related to SARS-CoV-2 (old section 2), making mention to BET proteins when appropriate.

Regarding section 5, in line with Reviewer comments, the section has been largely re-written. Although we keep 2 subsections, we have modified several aspects of both. In the first part, as Reviewer suggests, we have followed the sequence of SARS-CoV-2 infection to make mention to the different steps in which BET proteins should be implicated. We have also removed reference to processes involving BETs, but not directly related to SARS-CoV-2. In addition, we make mention to some of the aspects discussed in the second part when appropriate. In the second part, exploitation of BET inhibition for SARS-CoV-2 fighting is discussed in more detail, in light of the available recent reports. We indicate the model cells used in these studies and relate them to ACE2 and TMPRSS2 expression in organs. We sincerely apologize for forgetting to mention clinical trial NCT04894266; it has been included now.

Regarding discrepancies between text and figures, we have followed all the Reviewer comments on figures and corrected them for more clarity. Thus, regarding SARS-CoV-2 genome (now Fig. 2) we indicate that ORF for nsps is just 1 ORF (ORF1ab), which together with 4 ORFs for structural proteins and 9 ORFs for putative accessory proteins complete a total of 14 mentioned ORFs. In the same figure, part B, it is now specified that one element named membrane is the membrane (M) structural protein, while the other has been re-designated as virus envelope.  E protein position in relation to bromodomains has been also corrected on BET members scheme in the other figure, now designated as Fig. 1.

Reviewer 3 Report

This review by Nieves Lara-Ureña and Mario García-Domínguez provides a good summary of SARS-CoV-2 and related coronaviruses, covering a broad range of information from the clinical impact of the disease to the molecular functions of the SARS-CoV-2 genes and proteins.  The sections covering BET proteins provide a thorough review of this protein family covering a breadth and depth of knowledge that spans structure, molecular function, and involvement in human disease from cancer to viral pathogenesis.  Overall, the writing is strong and the composition is well done with minor corrections or clarifications needed in some passages.  The topic of BET family proteins in the context of SARS-CoV-2 is interesting and would be relevant and informative for the field.  Despite these strengths, the review fails to live up to its title and doesn’t do enough to synthesize the two main topics, namely BET proteins and SARS-CoV-2.  Too much time is spent keeping the two topics separated, with not enough attention spent on how they matter to each other.  In the end, it felt like I’d read a mini-review on SARS-CoV-2/COVID-19, followed by a very thorough review of the BET family of proteins and their functions in cells and disease.  Though each section is well done and overall the review flows nicely, revisions should be considered before the review is accepted.

Major Points:

  1. The introduction is almost entirely focused on SARS-CoV-2 and the epidemiological and clinical consequences of COVID-19. The information is accurate and interesting, but never gets directly tied back to the BET family of proteins.  This is a huge missed opportunity to synthesize the two topics right up front, either by connecting the symptoms and consequences of SARS-CoV-2 infection to host-directed therapy and potential use of BET inhibitors to treat COVID-19, or by connecting to BET protein involvement in inflammation and immunity.  There are a lot of reviews that cover general SARS-CoV-2 information.  The title of the review indicates that this isn’t another review about SARS-CoV-2, but specifically about BET proteins and how they are relevant to infection.  As such they should concisely describe general SARS-CoV-2/COVID-19 information, cite articles and reviews when necessary, and then move quickly to attacking the topic at hand.
  2. The overall structure of the review should be reconsidered. By separating the two main topics into segregated sections the authors spend too much time on details that don’t answer the question of why BET proteins are relevant to SARS-CoV-2 infection.  
    1. In section two, a lot of time is spent on SARS-CoV-2 genome structure and protein function. It is unclear how this is connected to BET proteins, or why the extent of the detail provided is necessary.  It is not until the end that there is a very short paragraph that introduces the idea of BRD2 and BRD4 as being connected to E protein.  A theoretical restructuring of the review might have specific sections on S and E and how each of these SARS-CoV-2 proteins is connected to BET proteins and/or BET inhibitors.  Avoid too many details that distract from the main point.
    2. In a similar way, section three on BET proteins is too long and takes up significant space not talking about a connection to SARS-CoV-2. The two subsections should be viewed through the lens of SARS-CoV-2 infection.  How do BET functions relate to coronavirus infection?  How does cancer tie back into infection (perhaps through the development of BET inhibitors and the repurposing of host directed therapy for treatment of virus infection)?  Some creative reorganization and editing of the manuscript could provide the answers to these questions.
    3. Section 4 is a good introduction to BET proteins and their functions during infection of other viruses. My main comments here are that it would be nice to tie all of this back to coronaviruses and/or the development of host-directed therapies, or to integrate it into earlier sections.
    4. Section 5 comes very late in the review, and could be integrated throughout. In section 5.1, the first paragraph is very long, compared to the sections that precede it.  A lot is covered in this paragraph, it would be good to have breaks to indicate new topics, or better yet, distribute this information into the sections that come before.
    5. The conclusion (section 6) of the review is very well-written and perfectly outlines how the review could be restructured. It seamlessly connects each main topic covered in the review, and ties each feature of BET proteins to SARS-CoV-2. 
  3. Spend more time describing the SARS-CoV-2 experiments in section 5.2. This should be the bulkier portion of the review and I was hoping for significant details here, especially since so much detail was provided in the other sections.  This section was more difficult to follow since certain details were missing.  A figure that synthesizes the information in this section would also be a great way to graphically help the reader understand what BET proteins are doing during infection with SARS-CoV-2.
  4. Consider updating or adding more information to the figures. While they are clean and easy to understand, as they are they don’t feel essential to the understanding of the manuscript.  In contrast, Table 1 is essential and provides a very nice summary of section 4.  

Minor Points:

  1. In a few cases, the language and word choice used should be altered to clarify content and make statement more accurate:
    1. Line 46 – 48; consider rewording “Replication of SARS-CoV-2 in the lungs provokes a strong immune response and ….”. As written, the sentence seems to imply that replication of SARS-CoV-2 in the lungs always provokes a strong immune response leading to a cytokine storm.
    2. Line 86: consider rewording to describe the proteolytic activity results in cleavage of the polyproteins into 16 non-structural proteins, some of which form the RTC.
    3. Line 97-98: This is either too much detail or not enough; it would be better to either remove this level of detail or provide more context for S1 and S2.
    4. Line 122: reword/adjust language. Though SARS-CoV and SARS-CoV-2 do use ACE2 as the receptor to infect cells, it’s a little bit of a stretch to state they must have the same life cycle from start to end.
  2. There are a number of minor language changes that would improve the flow or understandability of the manuscript.
    1. Line 51-52: reword sentence to improve flow
    2. Line 60: change “proteins of the SARS-CoV-2 have open new possibilities” to “proteins of SARS-CoV-2 opened new possibilities”
    3. Line 61-64: reword sentence to improve flow and understandability
    4. Line 85 and then again in line 87: avoid using “placed” to describe localization of genes in the genome
    5. Line 385: typo, “with acts specifically” I think should be “which acts specifically”
    6. Line 441: reword/change language “was able to avoid expression of key”; be more precise, is I-BET downregulating expression of these genes or is it regulating expression in response to something?
    7. Line 448-450: reword
    8. Line 452: it is unclear what the phrase “either on cell surface as intracellularly” is referring to, is this indicating the cell surface receptors act the same way as intracellular receptors?
    9. Line 456: reword to improve flow
    10. Line 525: reword for clarity, “Either BRD2 knockdown as BET inhibitors,”
    11. Line 528-531: reword for clarity
  3. Line 144-148: the references are missing here
  4. Line 409-415: the references are missing here

Author Response

Regarding major point 1: Despite this is a short introduction on severity of the recent pandemic, we have added some sentences in the last paragraph introducing BET relation with SARS-CoV-2. We have preferred to keep this section short, but instead, following Reviewer recommendation we immediately go through BET protein section (now section 2), where we make mention at several point to BET relation with SARS-CoV-2.

Regarding major point 2: We have now restructured and reordered the different sections. Former section 2 regarding virus structure is now part of section 3 on BET relation with viruses. Information on virus structure has been now shortened and more emphasis is put on E and S proteins as they are related to SARS-CoV-2. As explained above, former section 3 is now section 2, whit general information on BETs, which has been shortened and where we make mention to SARS-CoV-2 at several points. We think it is important to keep information on general functions of BETs, as this is relevant to understand why BET inhibitors have the potential to fight SARS-CoV-2. Next section, now section 3 (former section 4), is on BET relation with viruses, where we have shortened information related to classic viruses and introduced as a subsection the section on SARS-CoV-2, but referring to BET proteins when appropriate. Despite information on classic viruses and BETs is long, we think this is an opportunity to update the field, because the way BETs interact with classic viruses may be informative on the ways they interact with new viruses, which may shed light on the appropriate strategies to fight SARS-CoV-2. Regarding section 5, some of the discussed ideas have been now introduced before, but we think better to keep to the end detailed information on the use of BET inhibition for fighting SARS-CoV-2 as former sections contain the previous needed information to mechanistically understand the use of BET inhibitors. The first paragraph of former subsection 5.1 has now been largely reformulated and shortened by describing sequential steps in viral infection in relation to BET functions, removing mention to processes not directly related to SARS-CoV-2. Thus, we now think that new sections structure is better in line with conclusions. 

Regarding major point 3: We now spend more time in former section 5.2 (now 4.2), giving more details on recent reports related to the use of BET inhibitors against SARS-CoV-2. Following the Reviewer recommendation, we have generated an additional figure (Fig. 3) to illustrate the ways BET proteins are linked to SARS-CoV-2 infection.

Regarding major point 4: These figures together with Table-1 have been created just to schematize some of the introductory information on BET family, BET relation with viruses and SARS-CoV-2 structure. On this last we have indicated that S and E proteins interact with ACE2 and BET bromodomains, respectively. As mentioned above we have now incorporated an additional figure (Fig. 3) to summarize how BETs associate with SARS-CoV-2 infection.

Regarding minor point 1: All minor points have been addressed as indicated by the Reviewer. Regarding S1 and S2 subunits of S protein, we have decide to remove this information according to Reviewer suggestion.

Regarding minor point 2: All language changes suggested have been addressed.

Regarding minor point 3 and 4: We have introduced references where required. In relation to former lines 409-415, some unnecessary information has been now removed to shorten this section.

Round 2

Reviewer 1 Report

Thanks to the authors for the answer.

Author Response

We thank the Reviewer for this assessment

Reviewer 2 Report

Revisions applied to manuscript by Lara-Ureña and García-Domínguez resulted in dramatic improvements.  There is much more clarity and stronger connection in all sections to the title and objective “Relevance of BET family proteins in SARS-CoV-2 infection”.

Minor revisions are required to address unsupported statements (i.e. references required), typographical errors as well as incomplete or inaccurate statements.

1: Lines 159-162:

This statement introduces how gene expression for the host cell receptor responsible for SARS-CoV-2 is regulated by BET proteins, but does not indicate what the receptor is!  This leaves the reader hanging.  I recommend the authors (a) identify the receptor as ACE2 in this introduction in lines 159-162 and (b) refer to Section 3.2 where ACE2 involvement in COVID-19 is described.

2: Line 163: BET proteins play an important role in the control of cell proliferation.

This statement (a) appears to be out of place, as surrounding text is unrelated (b) does not include a supporting reference.  This sentence should be removed.

3: Line 265-267: Minor suggestion: This sentence is hard to read due to unnecessary placement of 2 commas that can be removed.

Original:

Thus, the development of drugs, able to displace BET proteins from the chromatin as a therapeutic approach to fight cancer, has been a highly active research objective in the last decade

Suggestion for authors:

Thus, the development of drugs able to displace BET proteins from the chromatin as a therapeutic approach to fight cancer has been a highly active research objective in the last decade

4: Line 271-273: Here the authors point out toxicity associated with BET inhibitors in clinical trials, and relate that to potential off-target effects.  This may be true, however, the section is incomplete. The authors should describe how the dose limiting toxicity of BET inhibitors in clinical trials is often on target (not off target) thrombocytopenia.  There are numerous publications on this including: doi 10.3389/fphar.2020.621093  doi 10.1038/d41573-020-00013-3

Further, bromoscan analysis indicates BET inhibitors that have been used in clinical trials have low affinity for bromodomains outside the BET family of proteins.  If the authors choose to keep the statement in lines 274-275 (which bromoscan suggests is inaccurate), then add the missing word “it”:  …making (it) difficult to completely discard off-target effects.

5:  Lines 353-354 and 776-778:  Here the authors describe BRD4 containing super enhancers (SEs).  In both sections, the authors indicate SEs are only involved in promoting disease.  As written, this is misleading.  The authors need to point out that SEs are present in the healthy state.  Only inappropriate assembly and/or dysregulation of SEs are associated with disease.  My point is that clearly SEs aren’t always “bad”.

Optional: It may be helpful for this review to emphasize the dynamic and often rapid formation of BET containing SEs in response to external stimuli, resulting in rapid and substantial changes to the transcriptome (Brown et al reference 217).  The authors elude to this in lines 783-790, but the dynamic nature of SEs and consequences of de novo SE assembly could be described better.

6:  Line 363: Heterozygous mice for these mutations also present defects, remarkably, reduced cell growth.

Fix bad English starting with “remarkably”

7:  Line 372: BET interaction with viruses is not restricted to SARS-CoV-2.

This is misleading as very few direct interactions between BET proteins and viruses have been identified.  The author may consider modifying to:

BET protein involvement in viral processes is not restricted to SARS-CoV-2.

8:  Lines 480-495 HIV section

This review should include the well described potential of BET inhibitors to reactivate HIV from latency for viral eradication.  Examples of literature describing this include: doi 10.1038/s41598-017-16816-1  10.3389/fmicb.2017.01035

9:  Lines 539-541:  Please fix bad English

10:  Lines 588-591   

This section describes SARS-CoV-2 variants of interest which have increased infectivity.  This section lacks mechanism.  The authors should include how increased infectivity of several SARS-CoV-2 variants can be related to (doi 10.1038/s41579-021-00573-0):

(a) mutations in viral Spike protein that have higher affinity for host ACE2

(b) mutations that reduce neutralizing activity of antibodies

An important consequence that the authors do not address is: because all variants of SARS-CoV-2 use ACE2 for entry into host cells, it’s predicted that strategies targeting ACE2 (including BET inhibitors) should decrease infection with any SARS-CoV-2 variant.  This concept should be included in this review.

11: Line 630:  Please review for a typographical error.

Should “BET-E protein-BET.” be “BET-E protein interaction.”?

12: Line 662: Typographical error: “especially” not “specially”.

13: Line 669-671:  The S protein of SARS-CoV-2 is likely recognized by TLR4, resulting in activation of the NF-KB pathway.

Reference required.

14: Line 673-674: SARS-CoV-2 may also activate the endosomal receptor TLR3, since this sensor is activated by foreign RNAs molecules (particularly dsRNAs) derived from virus genomes.

Reference required.

15: Line 764-767:  Besides triggering inflammation, respiratory viral infections are linked to oxidative stress of the epithelium, whose cells activate the transcription factor Nuclear factor erythroid-derived 2-Related Factor 2 (NRF2), which displays protection against oxidation and inflammation.

Reference required.

16: Lines 794-795:  This text appears to be dissociated from the previous paragraph.  Please correct.

17:  Line 858: The list of BET inhibitors that Mills et al demonstrated improve diastolic dysfunction modeled in hCO is incomplete.  This is important as this publication highlights the potential for BET inhibitors to reduce symptoms of “long COVID”, which is currently difficult to treat.

18: Line 883: Typographical error: “apabetalone” not “apabetolane”.

19: Line 904: Typographical error: “SSRP1” not “SSPR1”.  Also is “conform” a typo?  Should this be “can form”?

20: Line 904-906:  SUPT16H protein together with SSPR1 conform FACT (FAcilitates Chromatin Transcription), a heterodimeric histone chaperone associated with chromatin remodeling during gene transcription.

Reference required.

21: Line 949-950:  A number of defined viral proteins have been revealed to target BET proteins…

I do not believe this is true.  The statement is at least inconsistent with information in this review.  The authors only describe direct interaction between SARS-CoV-2 E-protein with host BRD2 and BRD4.  Do the authors mean BET proteins are involved in regulating a number of viral processes?

22:  Line 961:  Please fix bad English.

Reviewer 3 Report

The changes to the overall structure of the review are significant, and several portions have been re-written and re-organized.  Overall, the flow of the review is much improved and the two topics, SARS-CoV-2 and BET proteins, are much more integrated.  Despite the improvements, there are several changes that would be needed before I would recommend accepting the revision for publication.  Overall, the review needs moderate edits for clarity and language, particularly in the re-written sections to improve the understandability of the text.  In addition, section 3 and section 4 have large chunks of text that seem misplaced.  In section 3, which is about viruses in general, there is a section on SARS-CoV-2.  In section 4, which is on SARS-CoV-2 and BET proteins specifically, the bulk majority of section 4.1 seems to be about viruses in general (although this is not super clear in the text).  The following are suggestions and edits by section with line number corresponding to the clean version of the PDF:

  • Abstract
    • Ln 15-16: “The fundamental role BET proteins play in transcription constitutes the perfect focus for the propagation strategies of viruses.” What does this mean?
    • Ln 18: “… proteins from the chromatin, has boosted the interest on BETs…”
  • Introduction
    • Ln 39-44; Ln 51-57: I’m still not sure why we need to know quite this level of detail on the symptoms and consequences of COVID-19. It’s good to report that there are consequences, but the level of detail feels out of place for a review on BET proteins. This paragraph would be better served if the symptoms and consequences could be stated and wrapped up succinctly, then it should end on (or at least mention), what the current state of treatments and preventions is.  That would lead into why we are looking at host factors for druggable targets.
    • Ln 47: “associates” à associated
    • It would be useful to mention and define “host-directed therapy” somewhere in the last paragraph (rather than waiting to the next section to introduce this topic, it should come much earlier).
    • Ln 65: “…SARS-CoV-2 and the COVID-19…”
  • BET proteins
    • This overall flow of the review is much better, and the first introduction section does a better job integrating SARS-CoV-2. My main comment for the intro section is the overall need for more editing to make the language more precise and easier to follow.  Use of active rather than passive language would clean up a lot of this section. 
      • Ln 73: “providing new strategies to fight the virus.” This statement feels abrupt and out of place, mainly because host-directed therapy hasn’t been properly defined or introduced. This should be reworded, perhaps to indicate BRD2/4 are potential targets for host-directed therapy strategies. 
      • Ln 73-76: Reword; name the receptor and make the sentence clearer (e.g. “Moreover, several groups have demonstrated that ACE2, the SARS-CoV-2 receptor for host cell entry, is under BET protein transcriptional regulation.”).
      • Ln 93-95: Reword. (e.g. “Despite the high selectivity of BET inhibitors for BET bromodomains, 44 additional human proteins have bromodomains, making it difficult to completely discount off-target effects.”)
      • Ln 96-98: Reword and collapse into a single sentence. (e.g. “Though BET inhibitors have limited success in the treatment of some cancers, recent reports have shown BET inhibition as a promising strategy for treating SARS-CoV-2 infection.”
      • The last paragraph is not as clear as the proceeding paragraphs and lacks focus. I would either remove the statement on BET inhibitors treating other diseases, or move it to the paragraph above as a last sentence that introduces the idea of using BET inhibitors for treatment of other diseases.  Then the focus of the last paragraph could be on BET and viruses and using BET inhibitors to treat viral infection. 
    • The editing of the subsections has done a good job providing more precise descriptions, my one comment is that each section could still be connected back to SARS-CoV-2. Each section brings up useful information about BRD proteins and functions that could be linked back to SARS-CoV-2 or viruses, or at least introduce the concept.  It doesn’t have to be as detailed as the new sections 3 or 4, but mentioning these concepts throughout the text will ground the reader and help them understand how these functions could be important for the virus.
      • Section 2.1. BET structure. There is mention of BET proteins binding to H4 via acetylation of K5 and K12.  Is there any evidence that SARS-CoV-2 proteins interact with Histones or Histone Acetylation proteins?  Does SARS-CoV-2 infection alter transcriptional programs?  These types of things would be interesting to discuss and point out if known.
      • Section 2.2. General BET functions. While it is interesting to know about BET proteins functions outside viruses, linking some of these functions to viruses in general, or SARS-CoV-2 would be useful for the reader.  Do SARS-CoV-2 proteins interact with any cell cycle pathway proteins?  Is there any evidence for alterations to host cell cycle in SARS-CoV-2 infected cells?  A number of viruses can disrupt host cell cycle, is it known that BRD proteins are involved in this?  P-TEFb and HEXIM1 are involved in viral transcription programs, it would be interesting to mention that here. 
    • BET relation with viruses
      • 1 BET proteins and classic viruses.
        • What are defined as “classic viruses”?
        • This section is a little abrupt. An introduction would be useful, perhaps to describe that BET proteins are involved with a number of viruses in different ways. 
        • For readers unfamiliar with human papillomaviruses it might not be clear what “high-risk” and “low-risk” viruses are. Given the connection of HPV, KSHV, MHV, and EBV with cancer, and the connection of BET proteins and cancer, it might be good to connect viral replication, BET proteins, and cancer.
        • I would suggest combining all the herpesviruses paragraphs into a single paragraph and focus on LANA and LANA homologues since they are all functionally similar.
        • In the same way, connect all the retroviruses.
        • Table 1: Capitalize “Protein/Genome” to match the other labels, and indicate it is the viral protein/genome feature.
      • 2. The SARS-CoV-2.
        • This section might be better suited for section 4 since section 3 seemed to be more about viruses in general, and section 4 was more about SARS-CoV-2 specifically.
        • Remove “The” from title.
        • Ln 294-300. Reword for clarity.
        • The first paragraph does not flow well into the second paragraph. Perhaps mention of BET proteins connection to other coronaviruses would make that transition smoother.
        • Ln 343-347: Edit for clarity and content: “SARS-CoV uses the same receptor that SARS-CoV-2 uses to infect cells…”. Is binding stronger (by kd) or more likely (statistically)?  It would be good to clarify this a little more.
        • Ln 355: “The RTC complex…”
      • BET protein and SARS-CoV-2 infection
        • The overall introduction section needs editing and rewording for clarity, flow, and word choice.
        • 1 SARS-CoV-2 induced immune response and BET proteins
          • This section is pretty hard to follow and its unclear precisely what evidence demonstrates how SARS-CoV-2 induces either adaptive or innate immune response. In addition, it is unclear how BET proteins are involved in SARS-CoV-2 specific innate immune response, or if the authors are referring to general viral immune response.  If the latter is the case, these portions would feel better suited for Section 3.
          • The first paragraph is fairly speculative and/or missing references and in places it is unclear what is speculation and what is supported by evidence. It would be good to edit this paragraph to clarify these cases.
          • Ln 410: add comma “…and specifically BRD4, are primary….”
          • Ln 413-4: “The S protein of SARS-CoV-2 is likely recognized by TLR4, resulting in activation of the NF-kB pathway” Missing Reference.
          • Ln 422: missing ref
          • The second paragraph introductory statement is hard to follow, as it seems to imply that SARS-CoV-2 recognition (here I am assuming they mean immune recognition) converges on NFKB, though I’m not sure what this means. In addition, the paragraph that follows doesn’t really explain or follow-up this statement.  Instead it is a description of the NFKB pathway and BRD4, but not on SARS-CoV-2.  How does this connect to SARS-CoV-2?  Or is this more about BRD4 and viruses in general?
          • Similarly, in the third paragraph it is unclear if the authors are talking about SARS-CoV-2 infection or just viral infection in general. In addition, it is unclear the connection between TLR3 and SARS-CoV-2.  Is this speculative or supported by evidence.  If evidence-based, there need to be references supporting each statement.
        • 2 BET inhibition for COVID-19 treatment
          • Only minor comments and edits needed.
          • Ln 502: “…the virus infection, will result in…” change “will” to “may”
          • Ln 511-512: reword for clarity
          • Ln 537: Last word should be “proteins” not protein.
